# Interaction of Nucleolin with the Fusion Protein of Avian Metapneumovirus Subgroup C Contributes to Viral Replication

**DOI:** 10.3390/v14071402

**Published:** 2022-06-27

**Authors:** Dedong Wang, Lei Hou, Ning Zhu, Xiaoyu Yang, Jianwei Zhou, Yongqiu Cui, Jinshuo Guo, Xufei Feng, Jue Liu

**Affiliations:** 1College of Veterimary Medicine, Yangzhou University, Yangzhou 225009, China; wddyzu@163.com (D.W.); hlbj09@163.com (L.H.); zhuning7525@163.com (N.Z.); yxiaoyu0512@163.com (X.Y.); jwzhou@yzu.edu.cn (J.Z.); cuiyongqiu97@163.com (Y.C.); jane9407@163.com (J.G.); xffeng@yzu.edu.cn (X.F.); 2Jiangsu Co-Innovation Center for Prevention and Control of Important Animal Infectious Diseases and Zoonoses, Yangzhou University, Yangzhou 225009, China

**Keywords:** avian metapneumovirus subgroup C, fusion (F) protein, NCL, AS1411, replication

## Abstract

Avian metapneumovirus subgroup C (aMPV/C) is highly pathogenic to various avian species with acute respiratory tract clinicopathology and/or drops in egg production. Nucleolin (NCL), an important nucleolar protein, has been shown to regulate multiple viral replication and serve as a functional receptor for viral entry and internalization. Whether NCL is involved in aMPV/C pathogenesis is not known. In this study, we found that aMPV/C infection altered the subcellular localization of NCL in cultured cells. siRNA-targeted NCL resulted in a remarkable decline in aMPV/C replication in Vero cells. DF-1 cells showed a similar response after CRISPR/Cas9-mediated knock out of NCL during aMPV/C infection. Conversely, NCL overexpression significantly increased aMPV/C replication. Pretreatment with AS1411-a aptamer, a guanine (G)-rich oligonucleotide that forms four-stranded structures and competitively binding to NCL, decreased aMPV/C replication and viral titers in cultured cells. Additionally, we found that the aMPV/C fusion (F) protein specifically interacts with NCL through its central domain and that AS1411 disrupts this interaction, thus inhibiting viral replication. Taken together, these results reveal that the aMPV/C F protein interacts with NCL, which is employed by aMPV/C for efficient replication, thereby highlighting the strategic potential for control and therapy of aMPV/C infection.

## 1. Introduction

Avian metapneumovirus (aMPV), which belongs to the genus Metapneumovirus within the Paramyxoviridae family of viruses [1], can result in severe acute upper respiratory tract clinicopathology, egg-dropping, and poor egg quality in turkeys and chickens [2,3,4]. aMPV is a single-stranded, non-segmented, negative-sense RNA virus that contains 5′ and 3′ untranslated regions (UTRs) and eight transcription units (N, P, M, F, M2, SH, G, and L) [1,5]. Based on genetic characterization and antigenic features, aMPV can be classified into four subgroups: A, B, C, and D [6]. Subgroup C aMPV (aMPV/C) was first successfully isolated from turkeys in the United States, and subsequently, aMPV/C was successfully isolated from farm ducks, pheasants, and wild birds in France and South Korea [7,8,9]. We isolated aMPV/C from infected meat-type chickens with severe respiratory symptoms in southern China [7]. This is the first report of aMPV/C infection in chicken. It has been reported that aMPV/C participated in the regulation of multiple biological processes for viral replication [8,9].

Nucleolin (NCL), a highly conserved protein during evolution that is located in the nucleolus [10,11,12], participates in a variety of biological processes, such as chromatin organization and stability, ribosome biogenesis, apoptosis regulation, and stress response [13,14,15,16]. In addition, changes in subcellular localization are observed under various stimulations, such as viral infections [17,18,19]. Therefore, relocation of the NCL often indicates the occurrence of pathological phenomena in cells. Owing to its special shuttling property, the NCL acts as a bridge between homeostasis and external stimulation. An increasing number of studies have shown that NCL has an indispensable function in viral replication. For example, clathrin-dependent entry of rabbit hemorrhagic disease virus (RHDV) is regulated by NCL’s interaction with the RHDV capsid protein [20]. NCL is also associated with dengue virus (DENV) capsid protein, and this interaction can efficiently promote the production of DENV progeny and benefit DENV replication [21]. Apart from its positive role in viral replication, NCL has also been shown to serve as a receptor for some viruses like human immunodeficiency virus type 1 (HIV-1), human respiratory syncytial virus (RSV), and enterovirus 71 (EV71) [22,23,24].

In this study, we demonstrate that NCL translocates from the nucleus to the cytoplasm during aMPV/C infection and prompts the replication of aMPV/C. Both NCL knockdown and treatment with AS1411 dramatically inhibit aMPV/C replication. Additionally, aMPV/C F protein interacts directly with NCL in the cytoplasm. Notably, NCL specifically interacts with the aMPV/C F protein through its central domain (RBD), which plays a vital role in aMPV/C replication. Overall, our results demonstrate that the interaction of NCL with the F protein via its central domain RBD contributes to aMPV/C replication.

## 2. Materials and Methods

### 2.1. Cells and Virus Infection and Determination

Vero and DF-1 cells obtained from the American Type Culture Collection (ATCC)) were cultured at 37 °C in the presence of 5% CO_2_ in an incubator. They were maintained in Dulbecco’s modified Eagle’s medium (DMEM; Gibco, Life Technologies, New York, NY, USA) supplemented with 10% and 5% fetal bovine serum (FBS; Gibco, Life Technologies, New York, NY, USA), respectively, and 1% penicillin-streptomycin solution (Solarbio, Beijing, China, P1400). The aMPV/C strain JC was isolated from southern China [10] and stored in our laboratory. Both the cell lines were infected with this strain according to the following protocol. Cells were seeded on a six-well cell culture plate. After reaching 80% confluence, the cells were infected with aMPV/C at a multiplicity of infection (MOI) 1 and cultured in DMEM supplemented with 2% FBS in an incubator with 5% CO_2_ at 37 °C. These cells were harvested at the indicated times for virus titration, using the 50% tissue culture infective dose (TCID50) assay. Briefly, 10-fold serial dilutions of the harvested-cell supernatants were cultured in Vero or DF-1 monolayers in 96-well culture plates. Cytopathic effects (CPEs) were observed at 96 h after infection, and virus titers were expressed as TCID50 per 0.1 mL.

### 2.2. Plasmids Construction and NCL Expression

The Cas9-T2A-GFP and U6-sgRNA plasmid used for CRISPR/Cas9 technique in this study were kindly provided by Dr. Shihao Chen (Yangzhou University, Yangzhou, China). Three NCL-target single-guide RNAs (sgRNAs) were designed using the CHOPCHOP online design tools, and the recombinant U6-sgRNA plasmid was subsequently constructed. The oligo sequences of the two NCL-targeted single-guide RNAs are listed in Table 1. The cDNA of monkey NCL gene from Vero cells (GeneID: 103218056) was prepared by reverse transcription (RT)-PCR and subcloned into pCMV-HA (Clontech, TaKaRa Bio Inc, Ōtsu, Japan, 631604) to obtain the recombinant pCMV-HA-NCL plasmid. The full-length cDNA sequences of NCL and truncated NCL variants were amplified from Vero cells and cloned into vectors pEGFP-C1 using specific primers. The resultant plasmids were GFP-△NTD (318-772aa), GFP-△CTD (1-658aa), GFP-△NTD + △RBD (658-772aa), GFP-△NTD + △CTD (318-658aa), and GFP-△RBD + △CTD (1-318aa). Mutants were created by site-specific mutagenesis. Recombinant plasmids of the chicken NCL gene and aMPV/C-associated viral gene were generated in the same way as described above. These plasmids were sequenced to ensure that the PCR amplification did not introduce unwanted mutations or mismatches. Vero cells or DF-1 cells were seeded on a six-well cell culture plate, and when the cells grew to 80% confluence, the cells were transfected with these recombinant plasmids using Lipofectamine 2000 transfection reagent for further experiments.

### 2.3. Knockdown of NCL by RNA Interference (RNAi)

Three NCL-specific siRNAs were designed by the GengPharma Company (Suzhou, China). The sequences of the designed siRNAs are as follows: siNCL-A (sense, 5′-CCUUGGAACUCACUGGUUUT-3′; antisense, 5′-AAACCAGUGAGUUCCAAGGTT-3′), siNCL-B (sense, 5′-GCUUCAUUCGAAGAUGCUATT-3′; antisense, 5′-UAGCAUCUUC-GAAUGAAUGAAGCTT-3′), and siNCL-C (sense, 5′-GGUGGAAAGAAUAGCACUUT-T-3′; antisense, 5′-AAGUGCUAUUCUUUCCACCTT-3′). Vero cells were transfected with 20 μM siRNA using Lipofectamine RNAiMAX reagent in accordance with the manufacturer’s protocol and harvested for western blot analysis after 48 h. AS1411 (5′-GGTGGTGGTGGTTGTGGTGGTGGTGG-3′) and the negative control CRO (5′-CCTCCT-CCTCCTTCTCCTCCTCCTCC-3′) were obtained from Integrated DNA Technologies, Inc., Coralville, IA, USA, and resuspended in phosphate-buffered saline (PBS). Vero cells were treated with 4 μM AS1411, infected with the virus, or transfected with plasmids after 24 h. Before treating cells with AS1411, its toxicity was tested using the CCK-8 Cell Proliferation and Cytotoxicity Assay Kit (Solarbio, Beijing, China, CA1210), as per the following protocol. Vero cells or DF-1 cells in the logarithmic growth phase were seeded in a 96-well cell culture plate. Then, 4 μM of AS1411 was added after the cell density reached 70–80%. Following 72 h of AS1411 treatment, the supernatant was discarded and 90 μL of fresh cell culture medium and 10 μL of CCK-8 reagent were added to the matching cells. These were then cultured in an incubator for 3 h. Finally, Tecan infinite M Plex was used to measure absorbance at 450 nm.

### 2.4. Co-Immunoprecipitation (Co-IP) Analysis

Following transfection with the pCMV-HA-F plasmid, Vero cells were harvested after 36 h. Total cell lysates were extracted using NP-40 lysis buffer containing a 1 mM protease inhibitor cocktail (Beyotime, Shanghai, China, P1005). HEK293 cells were co-transfected with pCMV-HA-NCL and pEGFP-F plasmids, and total cell lysates were extracted after 36 h for Co-IP analysis using a commercial Co-IP kit (Thermo Scientific, Rockford, IL, USA) according to the manufacturer’s instructions. Protein A/G Plus Agarose was incubated with anti-HA monoclonal antibody (mAb) (Abcam, Cambridge, UK) or anti-GFP mAb (Sangon Biotech, Shanghai, China) for 12 h at 4 °C. The whole-cell lysate was incubated with this antibody-coupled resin for 4 h at 4 °C. It was then boiled in 4× SDS sample buffer and subjected to sodium dodecyl sulfate-polyacrylamide gel electrophoresis (SDS-PAGE), followed by immunoblot analysis.

### 2.5. Western Blotting

Protein extraction from whole cells was carried out using the RIPA lysis buffer (Beyotime, Shanghai, China, P0013B) with a 1 mM protease inhibitor cocktail (Beyotime, Shanghai, China, P1005) at the indicated times after transfection or infection. The cytoplasmic and nuclear protein extracts were prepared using Thermo Scientific™ NE-PER™ Nuclear and Cytoplasmic Extraction Reagents (Thermo Scientific, Rockford, IL, USA, 78833). Proteins were separated on SDS-PAGE and transferred to a nitrocellulose membrane. The membrane was blocked with 5% (*w*/*v*) nonfat milk in TBST buffer (150 mM NaCl, 20 mM Tris, and 0.1% Tween-20, pH 7.6) for 2 h at room temperature (RT). The membrane was incubated with different primary antibodies at 4 °C overnight. Following washes with TBST buffer, the membrane was incubated for 2 h at RT with either anti-rabbit (BBI, Sangon Biotech, Shanghai, China, cat# D110058) or anti-mouse (BBI, Sangon Biotech, Shanghai, China, D110087) or both secondary antibodies according to the primary antibodies used. A SuperSignal West Femto Substrate Trial Kit (Thermo Scientific, Rockford, IL, USA, 34096) was used to detect the secondary antibodies, and immunoreactive bands were visualized using a chemiluminescence apparatus (Protein-sample, Santa Clara, CA, USA). Primary antibodies used in this study were as follows: NCL monoclonal antibody(mAb) (Santa Cruz Biotechnology, Paso Robles, CA, USA), HA monoclonal antibody(mAb) (Abcam, Cambridge, UK), FLAG monoclonal antibody (BBI, Sangon Biotech, Shanghai, China, D191041), GFP monoclonal antibody (BBI, Sangon Biotech, Shanghai, China, D191040), aMPV/C-N monoclonal antibody (prepared in-house), GAPDH (BBI, Sangon Biotech, Shanghai, China,D110016), ACTIN (BBI, Sangon Biotech, Shanghai, China, D191048), and H3 (Proteintech, Chicago, IL, USA, 17168-1-AP).

### 2.6. Quantitative Real-Time Reverse Transcription PCR

Two-step real-time RT-PCR was used to detect the mRNA levels of NCL, aMPV/C F, and GAPDH in Vero and DF-1 cells. The primers used in the experiment are listed in Table 2. RNA was isolated from whole cells using TRIzol reagent, and cDNA synthesis was performed using the Vazyme cDNA Synthesis Kit (Vazyme, Nanjing, China). All cDNA obtained from different samples was further analyzed using the AceQTM Universal SYBR qPCR Master Mix Kit (Vazyme, Nanjing, China) and the Roche LightCycler^®^ Real-time PCR detection system. The total 20 μL PCR reaction volume consisted of 10 μL of AceQTM Universal SYBR qPCR Master Mix, 0.5 μL of the cDNA, 9 μL of DNase/RNase-free H_2_O, and 0.4 μL of the forward and reverse primer each (the final concentration of each primer is 0.2 μM). The PCR conditions were as follows: 95 °C for 5 min, followed by 40 cycles of 10 s at 95 °C and 30 s at 60 °C. The 2^−∆∆Ct^ method was applied to calculate the relative expression levels of the target gene mRNAs, and all mRNA levels were normalized to the GAPDH gene.

### 2.7. Confocal Microscopy

Vero cells were seeded on a glass-bottom cell culture dish (Biosharp, BS-15-GJM), and when the cell density increased approximately up to 30–40%, the pCMV-HA-F plasmid was transfected into cells with the Lipofectamine 2000 transfection reagent (Invitrogen, Caribised, CA, USA, 11668019). Cells transfected with pCMV-HA plasmid acted as the negative control. Then, 36 h after transfection, the cells were fixed with 4% paraformaldehyde (Solarbio, Beijing, China, P1110) for 20 min at RT and permeabilized using 0.1% Triton (Solarbio, Beijing, China, P1080) for 10 min at RT. The cells were then blocked with 2% bovine serum albumin (BSA) (Beyotime, ST023), and incubated with a mouse monoclonal antibody against NCL for 4 h at 37 °C. After washing cells with PBS for 20 min, they were incubated with Alexa Fluor 546 goat anti-mouse (Invitrogen, Caribised, CA, USA, A11010) for 30 min at 37 °C and stained with DAPI (2,4-diamidino-2-phenylindole) (Sigma-Aldrich, NJ, USA, D9542) for 5 min at RT. A confocal immunofluorescence microscope (Leica TCS SP8 STED, Weztlar, Germany) was used to image these cells.

### 2.8. Statistical Analysis

Statistical analysis was performed using GraphPad Prism version 8.0 (GraphPad software, La Jolla CA, USA). Data were collected from three independent experiments and evaluated by the Student’s *t*-test for two groups. Differences were considered statistically significant at *p* < 0.05. The values are expressed as mean ± standard deviation (SD).

## 3. Results

### 3.1. aMPV/C Infection Induces Cytoplasmic Re-Localization of NCL from the Nucleus

Given that NCL has previously been reported to participate in the replication of multiple viruses, we first explored the effect of aMPV/C infection on NCL expression. Here, qRT-PCR and western blot analyses were used to detect the expression levels of NCL in aMPV/C-infected Vero cells over a 48 h time course. As shown in Figure 1A,B, the transcriptional and translational levels of NCL were not affected by aMPV/C infection. However, NCL was found to be enriched in the cytoplasm instead of the nucleus (Figure 1B). Taken together, these results showed that aMPV/C infection did not affect NCL expression; however, it induced cytoplasmic relocation of NCL from the nucleus.

### 3.2. NCL Promotes aMPV/C Replication

In order to determine the influence of NCL knockdown on viral replication, three siRNAs targeting NCL and a negative control (siNC) were utilized. Western blot analysis showed that 10 pmol siRNA had the highest knockdown efficiency compared to siNC-treated cells, and upregulation of aMPV/C replication was simultaneously observed (Figure 2A,C). Meanwhile, we treated Vero cells with pCMV-HA-N-NCL to evaluate the effect of NCL overexpression on viral replication, with pCMV-HA-N as a negative control (Figure 2B). This was found to reduce aMPV/C replication (Figure 2D). Furthermore, a reduction in aMPV/C replication in NCL-knockdown cells and an increase in aMPV/C replication in NCL-overexpressing cells as compared to that in aMPV/C alone-infected cells was observed. The above results indicate that NCL promotes aMPV/C replication.

### 3.3. Knockingout of NCL Limits aMPV/C Replication

To further explore the role of NCL in aMPV/C replication, NCL-knockout (NCL-KO) DF-1 cells were generated using the CRISPR/Cas9 system (Figure 3A,B). These cells were infected with aMPV/C for 48 h. Western blot analysis verified the complete loss of NCL in the NCL-KO DF-1 cells compared to wild-type cells (Figure 3B). Western blot analysis and the viral titer assay suggested that blocking the expression of NCL significantly inhibited viral protein expression and decreased viral production (Figure 3C,D). Moreover, DF-1 cells transfected with pCMV-HA-N-NCL to over-express NCL showed increased aMPV/C N protein expression compared to pCMV-HA-N-transfected cells (Figure 3E). Additionally, DF-1-KO cells transfected with pCMV-HA-NCL showed increased expression of aMPV N protein in the immunoblot analysis (Figure 3F).

### 3.4. aMPV/C Fusion Protein (F) Binds to NCL

An increasing number of studies have shown that NCL participates in the replication of multiple viruses through various mechanisms [25,26,27]. By binding its fusion glycoprotein to NCL, the human respiratory syncytial virus (hRSV) of the Paramyxoviridae family has been shown to use NCL as a receptor for viral replication [24]. Therefore, we investigated whether the fusion (F) protein of aMPV/C, also from this family, can specifically bind to NCL. An immunoprecipitation (IP) assay conducted on whole-cell lysates using NCL mAb indicated that aMPV/C F can interact with endogenous NCL (Figure 4A). To further determine whether NCL directly interacts with aMPV/C F, Co-IP was performed with an HA mAb with lysates of HEK293 cells, 36 h after co-transfection with pCMV-HA-NCL and pEGFP-C1-F plasmids. The assay revealed a direct interaction between aMPV/C F and NCL (Figure 4B). Confocal microscopy showed that aMPV/C F not only interacts with NCL but also causes re-localization of NCL from the nucleus to the cytoplasm (Figure 4C).

### 3.5. NCL-Binding Aptamer AS1411 Inhibits Viral Replication by Disrupting the Interaction between NCL and aMPV/C F

AS1411, a guanine (G)-rich oligonucleotide that can form four-stranded structures, has been used as an antitumor agent because it can selectively bind to NCL and intervene in NCL-mediated biological functions [21,22]. Because of this interaction, AS1411 also interferes with the role of NCL in multiple viral replications, thereby affecting virus replication [21,22,28]. We discovered that NCL promotes viral replication based on the above findings; therefore, we investigated the effect on aMPV/C replication when NCL’s function was competitively hindered. Prior to this, we verified that no evident cell death occurred upon treating Vero cells and DF-1 cells with AS1411 or CRO, a negative control sequence for AS1411, at a concentration of 4 μM for 72 h, by the CCK-8 assay (Figure 5A,B). Therefore, to assess the effect of AS1411 on viral replication, we pre-treated Vero cells and DF-1 cells with 4 μM of AS1411 or CRO for 24 h and inoculated them with aMPV/C. As expected, a significant decrease in aMPV/C viral protein expression and viral titers was observed (Figure 5C,D). These results confirm that NCL expression levels are positively correlated with aMPV/C replication and that AS1411 suppresses viral replication. Since AS1411 is well known for its binding to NCL, we assessed whether it suppressed viral replication by altering the expression of NCL. After 72 h of treating Vero cells with AS1411 or CRO, western blot analysis showed that there was no detectable impact on the expression of NCL (Figure 5E). We then investigated whether AS1411 had a direct impact on viral replication. To this end, we knocked down NCL expression by transfecting Vero cells with siRNA targeting NCL. After 24 h, these cells were treated with AS1411 or CRO for 24 h, followed by infection with aMPV/C for 48 h. Western blotting analysis indicated no effect on the viral N protein expression (Figure 5F). Given that AS1411 does not alter NCL expression or directly influence viral replication, we hypothesized that AS1411 inhibits viral replication by competing with aMPV/C for binding to NCL. IP tests revealed that AS1411 could inhibit the interaction of the aMPV/C F protein with NCL (Figure 5G).

### 3.6. The Central Domain of NCL Plays a Critical Role in Viral Replication

NCL contains three functional domains that mediate various biological processes and pathologies through distinct mechanisms [14,16,20]. To determine the crucial domain of NCL that facilitates viral replication, eukaryotic expression plasmids were generated containing sequences encoding either the full-length NCL or NCL deficient of distinct functional domains (Figure 6A,B). Vero cells transfected with these plasmids were infected with aMPV/C. Western blotting was used for further analysis. Both full-length NCL and the RBD of NCL facilitated viral replication (Figure 6C), confirming that the RBD is critical for aMPV/C replication. Plasmids expressing NTD and RBD of NCL (GFP△CTD) and plasmids expressing RBD and CTD of NCL (GFP△NTD) were constructed. Cells transfected with GFP△CTD plasmids were found to be more conducive to viral replication than cells transfected with CTD plasmids (GFP△NTD + △RBD). Furthermore, viral replication was higher in cells transfected with GFP-△NTD plasmids compared to that in cells transfected with NTD plasmids (GFP△RBD + △CTD) (Figure 6D,E). Given the importance of the RBD of NCL in viral replication, we performed an exogenous Co-IP assay to further explore whether the RBD interacts with aMPV/C F. RBD and full-length NCL were found to directly and specifically interact with aMPV/C F (Figure 6F).

## 4. Discussion

aMPV/C is a pathogen that causes severe respiratory disease and swollen head syndrome in infected chickens [4]. Since its discovery in the United States in 1996, aMPV/C has posed a serious threat to the health and long-term development of the global poultry industry, resulting in significant financial losses [2]. Numerous studies have shown that viruses replicate in host cells by utilizing the host signal pathways and/or proteins. A recent report of iTRAQ quantitative proteomics showed that multiple signaling pathways, such as mTOR, MAPK, and p53, participate in aMPV/C replication [9]. In this study, a series of experiments were carried out to determine the influence of NCL on viral replication. NCL influences the replication of multiple viruses, including members of the paramyxoviridae family, through various molecular pathways [21,24,25,28]. However, few studies have been published to date that demonstrate the function of NCL in aMPV/C replication.

NCL expression was not altered by the infection (Figure 1A,B). However, the subcellular location of NCL was affected. Although it has been reported that 90% of NCL in cells is localized in the nucleoli, the nucleolar-cytoplasmic redistribution of NCL has been linked to a variety of pathological events, including cancer and viral infection [29,30,31,32]. Additionally, nucleolar-cytoplasmic re-localization of NCL was first observed in poliovirus-infected cells, and dynamic subcellular localization of NCL was presented in adenovirus-infected cells [32,33]. Our data corroborate previous findings that viral infection is a cause of NCL re-localization. Interestingly, although the expression level of NCL was unaffected by aMPV/C infection, the replication level of aMPV/C was altered when NCL expression was artificially modified (Figure 2D,E), indicating that NCL might be involved in aMPV/C replication. These findings are consistent with previous studies on NCL using other viral species, including enterovirus 71 (EV71) [23], dengue virus (DENV) [21], and rabbit hemorrhagic sickness virus (RHDV) [20]. Additionally, NCL was knocked-out in DF-1 cells to further verify your findings (Figure 3A), as observed for the mechanisms through which NCL influences different viral replications [25,26,34]. When we examined the effect of NCL loss on viral replication, we discovered that it drastically reduced viral protein production and titer (Figure 3B,D). In summary, our findings demonstrate that aMPV/C infection mediates nucleolar-cytoplasmic re-localization of NCL and that it is essential for aMPV/C replication.

Previous studies have implicated the interaction between NCL and viral proteins as a key factor in virus-induced nucleolar-cytoplasmic re-localization of NCL. For example, nucleolar NCL accumulates in the cytoplasm over the course of human cytomegalovirus infection as UL84 interacts with NCL [35]. The Dengue virus capsid protein (DENV C protein) directly binds to NCL to facilitate viral infection. Employing AS1411 disrupts the DENV C and NCL interaction followed by decreasing viral titer [21]. In this study, we found that the nucleolar-cytoplasmic re-localization of NCL was induced by aMPV/C F (Figure 4C) and that NCL can directly interact with aMPV/C F (Figure 4A,B). This provides more definitive evidence that the nucleolar-cytoplasmic re-localization of NCL is dependent on the relationship between NCL and viral proteins. AS1411, an aptamer of NCL, has been suggested to prevent an increase in the viral protein expression induced by NCL [21,22]. Similarly, we found that aMPV/C replication was inhibited after treatment with AS1411 (Figure 5C,D). AS1411 binds to NCL, implying a viral replication-blocking mechanism mediated by AS1411 interference [36]. Previous research has suggested that AS1411 may also impair the assembly of the complex formed by a virion with the host protein [37,38]. Our data indicate that AS1411 hinders aMPV/C replication by inhibiting the interaction between NCL and aMPV/C F rather than influencing the expression of NCL (Figure 5E,G). This demonstrates that NCL is involved in viral replication through its interaction with viral proteins.

In addition to the nucleolar-cytoplasmic re-localization, many studies have also shown an increased expression of NCL on the cell surface in several cases of virus infection [39,40]. Although the process by which NCL translocates to the plasma membrane is unknown, NCL on the cell surface is critical for viral attachment and entry into the host [20]. It acts as a cellular receptor to facilitate the entry of human respiratory syncytial virus (hRSV), and also serves as a novel receptor to mediate the binding and infection of enterovirus 71 (EV71) [23,24]. Meanwhile, AS1411 can compete with viral proteins to bind to the NCL and significantly hinder the internalization of human immunodeficiency virus type 1 (HIV-1) [22]. Multiple influenza A viruses, such as H1N1, H3N2, H5N1, and H7N9, require cell surface NCL to enter the cells [34,41]. Among the eight structural proteins of aMPV, M, F, and G have been shown to participate in initial viral attachment and entry [42,43,44,45]; however, aMPV M is mainly associated with the inner surface of the membrane of infected cells [1]. Although our work shows that NCL’s subcellular location changed after viral infection, it is unclear whether NCL aggregates on the membrane after aMPV infection, which is an issue that needs to be addressed in the future. Based on the preliminary research from our lab, we found that viral replication was reduced in infected cells following treatment with NCL antibodies (data not shown). This indicates that NCL may serve as a receptor to regulate the attachment and entry of aMPV/C. While our data established that AS1411 can impede aMPV/C replication by inhibiting the interaction between NCL and aMPV/C F (Figure 5G), whether it affects the attachment and internalization of aMPV/C as shown previously for other viruses [22] is yet to be investigated further.

In addition to its unique feature shuttle, NCL is well known for its multifunctionality. The multifunctionality of NCL stems mostly from its multidomain structure. There are three main domains of NCL: the N-terminal domain, the central domain, and the C-terminal domain [14]. The N-terminal domain of NCL has been found to contain phosphorylation sites of multiple proteins involved in cell cycle regulation [46,47]. The central domain of NCL is known as the RNA-binding domain (RBD) because it contains four RNA recognition motifs [48,49]. RBDs are frequently involved in the precise folding of pre-rRNA, RNA packing, and mRNA stability [50,51,52]. RBDs are also involved in the nucleolar localization of NCL [53,54]. The C-terminal domain is also called the RGG domain, named after the numerous Arg-Gly-Gly (RGG) repeats present here [14,55,56]. According to a structural study, the RGG domain is required for the nuclear import of ribosomal proteins. It assists in an efficient binding between NCL and large RNA [19,57]. These three functional domains of NCL have previously been identified to interact with three distinct viruses respectively. Rabbit hemorrhagic disease virus (RHDV) VP60 binds to the N-terminal domain of NCL [20], while the C-terminal domain of NCL is required for its interaction with peste des petits ruminants virus (PPRV) N protein [25]. The RNA-binding domain of NCL is sufficient for NCL’s interaction with the FMDV IRES [58]. In this study, RBD was found to promote aMPV/C replication (Figure 6A–C) and bind to aMPV/C F (Figure 6C,F).

In conclusion, our findings show for the first time that NCL is critical for aMPV/C replication. The fusion protein of aMPV/C specifically binds to NCL through its central domain, which is utilized by aMPV/C for efficient replication. In addition to establishing NCL as an important cellular protein involved in aMPV/C replication, these findings underscore its strategic potential for the control and treatment of aMPV/C infection.

## Figures and Tables

**Figure 1 viruses-14-01402-f001:**
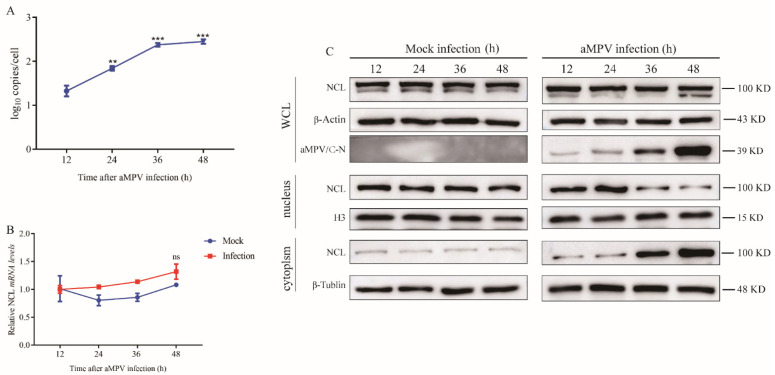
aMPV/C infection-induced cytoplasmic re-localization of NCL from the nucleus. Vero cells were infected or not with aMPV/C, and samples were harvested at 12, 24, 36, and 48 h following infection or non-infection. (**A**) aMPV/C RNA loads are displayed according to the number of hours following infection with aMPV/C, represented in copies/mL. (**B**) qRT-PCR analysis was used to identify relative quantities of NCL RNA in different samples obtained at the indicated time. (**C**) Western blot analysis was utilized to determine the total level of NCL expression, as well as the cytoplasmic and nuclear abundance of different samples obtained at the specified times. H3, β-actin, or β-tublin was used as an internal control. All data from three different experiments were shown as mean SD. ** *p* < 0.01, *** *p* < 0.001 were considered statistically significant; ns, not significant.

**Figure 2 viruses-14-01402-f002:**
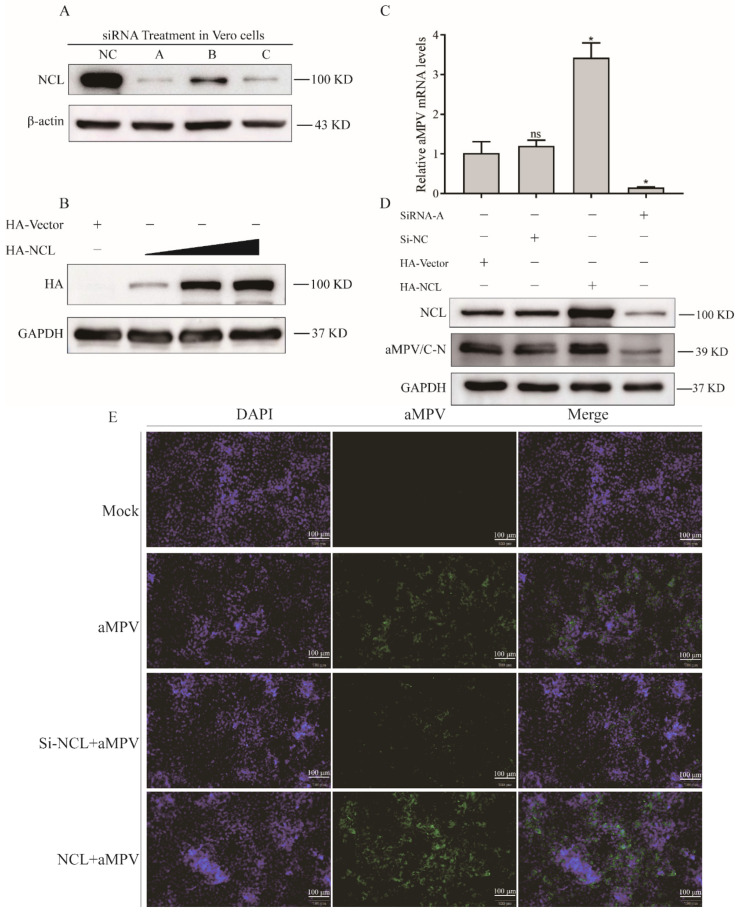
NCL promotes aMPV/C replication. (**A**,**B**) 20 μM of NCL−siRNA or increased amounts of pCMV−HA−N−NCL plasmids were transfected or mock−transfected into Vero cells. At 24 h, NCL proteins were detected by western blot. β−actin served as an internal control. (**C**) qRT−PCR was performed to determine the relative aMPV/C RNA level in Vero cells under both knockdown and overexpression of NCL, and western blot * *p* < 0.05 were considered statistically significant; ns, not significant. (**D**) was used to quantify the viral protein expression level under these conditions. The expression of total GAPDH protein and GAPDH mRNA were utilized as controls. (**E**) Vero cells transfected with NCL−siRNA or CMV−HA−N−NCL plasmids were infected with aMPV/C for 48 h, the cells were then directly observed under an immunofluorescence microscope and a specific antibody to the aMPV/C (green), and nuclei were stained with DAPI (blue).

**Figure 3 viruses-14-01402-f003:**
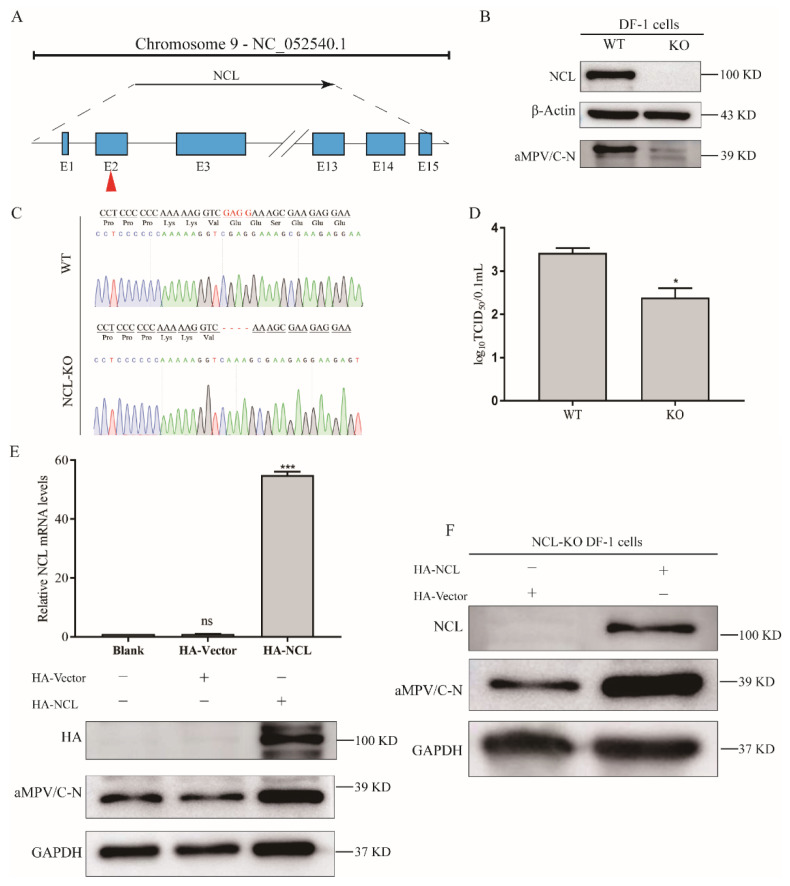
Knocking−out NCL limits the aMPV/C replication. (**A**) The genomic structure of chicken NCL gene and the knock−out strategy for NCL knockout using CRISPR/Cas9. NCL−KO cells and wild−type (WT) cells were infected with aMPV/C, and the cells and supernatants were collected 48 h later. Western blotting, sequence analysis, and TCID_50_ assay were used to evaluate the expression of NCL and aMPV/C N (**B**), DNA sequence (**C**), and the viral titer (**D**). NCL−KO cells and WT cells transfected with pCMV−HA−NCL plasmids were infected with aMPV/C, and the cells were harvested 48 h later for analysis of the relative NCL RNA level (**E**) and of the viral protein expression level (**F**) using qRT−PCR and western blotting. β−Actin or GAPDH was used as an internal control.

**Figure 4 viruses-14-01402-f004:**
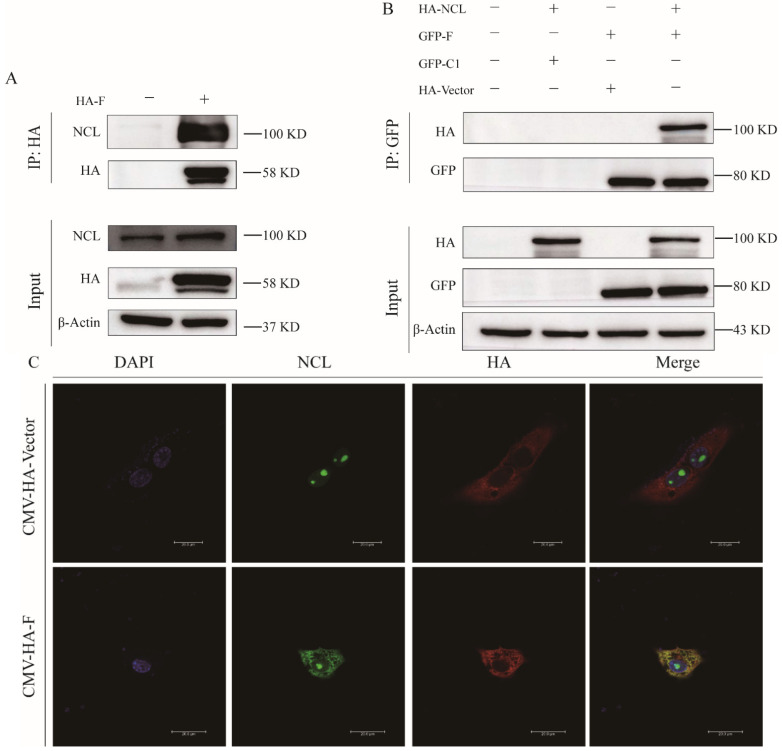
aMPV/C F protein interacts with NCL. (**A**) Vero cells were transfected or not transfected with the pCMV−HA−F plasmids for 36 h and an IP assay on cell lysates was performed using HA mAb, followed by immunoblotting with mAbs against HA and NCL. (**B**) The plasmids of pCMV−HA−NCL and pEGFP−C1−F were co−transfected into HEK293 cells for 36 h, a Co−IP assay was subsequently performed using GFP mAb, followed by immunoblotting with mAbs against HA and GFP. (**C**) Vero cells transfected or not transfected with pCMV−HA−F were incubated with NCL antibody (green), HA antibody (red), and DAPI (blue), and then observed by confocal microscopy. Scale bar, 200 μm.

**Figure 5 viruses-14-01402-f005:**
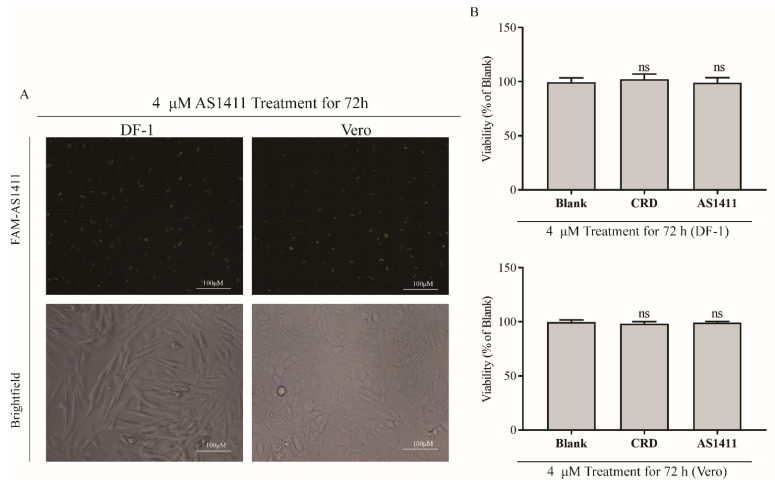
NCL binding aptamer AS1411 inhibits viral replication by disrupting the interaction between NCL and aMPV/C F. After 72 h of treatment with FAM−AS1411 in DF−1 cells and Vero cells, the FAM−signal and the toxicity of AS1411 were tested by confocal microscopy (**A**) and the CCK−8 assay (**B**). Vero cells and DF−1 cells treated with AS1411 or CRO for 24 h were infected with aMPV/C, and the cells and supernatants were collected at 48 h after infection. The viral titers (**C**) and expression of aMPV/C N and NCL (**D**) were assessed using western blot, qRT−PCR, and TCID_50_ assay. (**E**) After treatment of cells with AS1411 or CRO for 72 h, western blot was utilized to detect the expression of NCL in Vero cells. (**F**) Vero cells were treated with NCL−siRNA for 24 h and then infected aMPV/C in the presence of AS1411 or CRO. Western blot analysis was used to detect the expression of aMPV/C N and NCL. (**G**) Vero cells were transfected with pCMV−HA−F for 36 h in the presence of AS1411 or CRO. The cell lysates were extracted, and an IP assay was performed using HA mAb, followed by immunoblotting with mAbs against HA and NCL. β−actin or GAPDH was used as internal control. *** *p* < 0.05 were considered statistically significant; ns, not significant.

**Figure 6 viruses-14-01402-f006:**
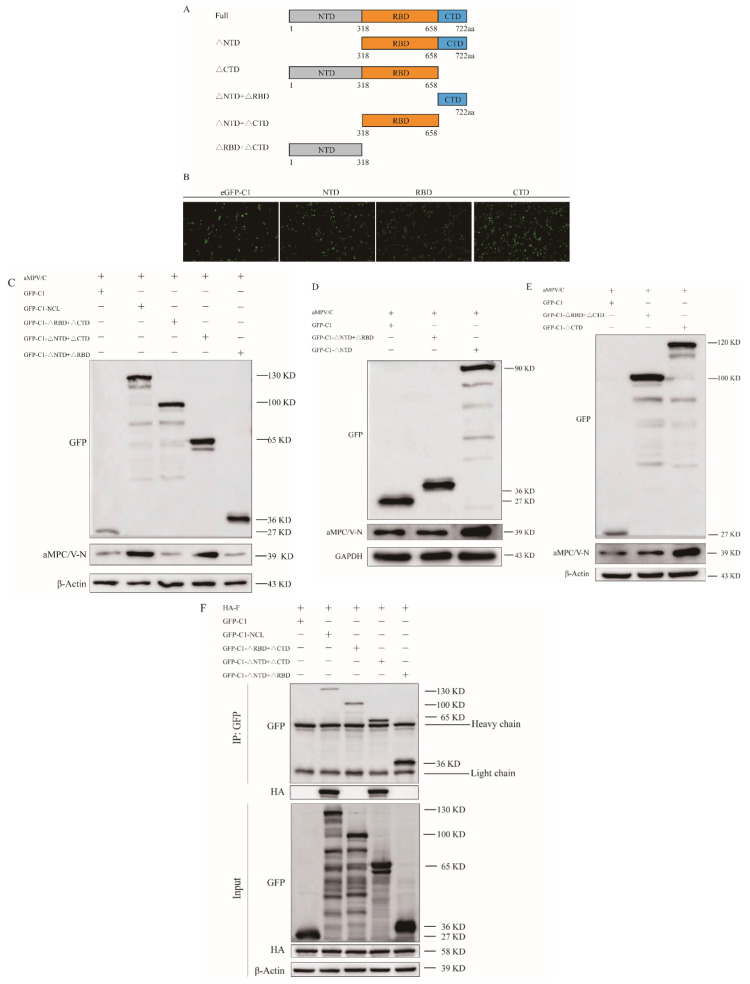
The central domain of NCL plays a critical role in viral replication. (**A**) The structural representation for constructing full−length GFP−tag NCL and NTD, RBD, CTD, NTD−deficiency, and CTD−deficiency domains of NCL. (**B**) After 24 h of transfection with GFP−tag NTD, RBD, and CTD domains of NCL in Vero cells, the GFP−signal was tested by confocal microscopy. Scale bar, 100 μm (**C**–**E**) Vero cells transfected with GFP−NCL with the indicated truncations were infected with aMPV/C for 48 h for analysis of aMPV/C N expression in western blotting. (**F**) pCMV−HA−F and GFP−NCL with the indicated truncations were co−transfected into HEK293 cells, a Co−IP assay was performed on cell lysates using GFP mAb, followed by immunoblotting with mAbs against HA and GFP. β−actin or GAPDH was used as an internal control.

**Table 1 viruses-14-01402-t001:** Oligonucleotide sequences (5′-3′) used for CRISPR/Cas9.

Name	Forward	Reverse
chNCL-gRNA1	cacc GTCCGACTTAGAGGAAAGCAG	aaac CTGCTTTCCTCTAAGTCGGAC
chNCL-gRNA2	cacc GCTTTCCTCGACCTTTTTGGG	aaac CCCAAAAAGGTCGAGGAAAG

**Table 2 viruses-14-01402-t002:** Primers (5′-3′) used in qRT-PCR.

Name	Forward	Reverse
GAPDH-Gallus	TAAGCGTGTTATCATCTC	GGGACTTGTCATATTTCT
GAPDH-Chlorocebus	CGGATTTGGTCGTATTGG	GTGGAATCATACTGGAACAT
NCL-Gallus	GACTATGAAGAACTGAGGACTG	CGCTTGGAAGAACCGATT
NCL-Chlorocebus	GACATCCACAACAGCAAGA	AAGGCACAGAACCGACTA
aMPV-M200	GTCAATTCAGCCAAGGCAGT	GGGGCAATCCTAGCTTGAGT

## Data Availability

Data are contained within the article.

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
