# Peer review of "Interaction of Nucleolin with the Fusion Protein of Avian Metapneumovirus Subgroup C Contributes to Viral Replication"

_viruses, 2022, doi:10.3390/v14071402_

Round 1
Reviewer 1 Report
In this manuscript, Wang and colleagues demonstrated that aMPV/C F protein specifically interacts with NCL protein, which promotes the replication of virus. The experimental design and data of the manuscript is of high integrity and quality, except for a little problem. For example, the clarity and brightness of fluorescent image may need to be improved. In figure 4B, a GFP-C1 need to be added as a empty control? The size of GFP protein is 21 KD or 27 KD?......
Author Response
Dear editor and reviewers,
Thank you very much for the chance you gave us to revise our manuscript. We thank the reviewers for the time and effort that they have put into reviewing the previous version of the manuscript. We admire your expertise and patience. The comments are very useful for our research and publication. We decide to accept all the comments and revise our manuscript carefully according to your comments. The changes were listed blew point by point, and were marked in red in the revised manuscript. The manuscript has greatly benefited from these invaluable suggestions.
Based on the instructions provided in your letter, we uploaded the file of the revised manuscript. Appended to this letter is our point-by-point response to the comments raised by the reviewers.
Reviewer 1
In this manuscript, Wang and colleagues demonstrated that aMPV/C F protein specifically interacts with NCL protein, which promotes the replication of virus. The experimental design and data of the manuscript is of high integrity and quality, except for a little problem. For example, the clarity and brightness of fluorescent image may need to be improved. In figure 4B, a GFP-C1 need to be added as a empty control? The size of GFP protein is 21 KD or 27 KD?......
As suggested by the reviewer, we have increased clarity and brightness of fluorescent image in Figure 2E of the revised manuscript.
We had really utilized a GFP-C1 as an empty control in IP experiments and the modification of the labels has been made in Figure 4B of the revised manuscript.
Regarding the size of the GFP protein, it is 27 KD in size after checking the raw data. We have corrected these errors and the relevant modifications have been made on Figure 6C, 6D, 6E, and 6F of the revised manuscript.

Reviewer 2 Report
This manuscript submitted by wang et al identifies the role of Nucleolin (NCL) in Avian metapneumovirus subgroup C (aMPV/C) infection. Using knockdown, CRISPR mediated knockout of NCL, the authors show that NCL is required for proper aMPV/C infection. Whereas overexpression of NCL in cell culture system facilitates aMPV/C infection. Immunoprecipitation of NCL identifies Fusion protein is interacting with NCL through the central domain and is critical for viral replication. Finally, the authors show that the inhibitor of NCL (AS1411-a aptamer) inhibit the interaction of NCL with Fusion protein and thereby reduces viral replication.
Overall, this study clarifies the role of NCL during aMPV/C infection. This is a well written manuscript, the experiments are done with proper controls, the results are clear, and the conclusions justifies the results. There are few minor errors in the current version of the manuscript.
1. Fig 2E – scale bar is needed as it is currently not clear what is the magnification range of the microscopic images.
2. Fig 3F – the label is bit confusing here. DF-1-KO-NCL looks like it represents NCL KO
3. Fig 6. The whole image panels need to be rearranged and fit into one page or make the figure panels smaller. It is currently hard as it spreads to three pages.
4. Fig 6A – The labeling for the domain needs to be clear. For example, ΔNTD+CTD seems like the construct has ΔNTD with CTD. This can be rectified by saying ΔNTD+ΔCTD or Δ(NTD+CTD).
Author Response
Dear editor and reviewers,
Thank you very much for the chance you gave us to revise our manuscript. We thank the reviewers for the time and effort that they have put into reviewing the previous version of the manuscript. We admire your expertise and patience. The comments are very useful for our research and publication. We decide to accept all the comments and revise our manuscript carefully according to your comments. The changes were listed blew point by point, and were marked in red in the revised manuscript. The manuscript has greatly benefited from these invaluable suggestions.
Based on the instructions provided in your letter, we uploaded the file of the revised manuscript. Appended to this letter is our point-by-point response to the comments raised by the reviewers.
Reviewer 2
Fig 2E – scale bar is needed as it is currently not clear what is the magnification range of the microscopic images.
As suggested by the reviewer, the clearer scale bars have been shown in all the microscopic images of Fig 2E of the revised manuscript.
Fig 3F – the label is bit confusing here. DF-1-KO-NCL looks like it represents NCL KO
As suggested by the reviewer, the label DF-1-KO-NCL has been modified to a new label NCL-KO (namely NCL KO) DF-1 in Fig 3F of the revised manuscript.
Fig 6. The whole image panels need to be rearranged and fit into one page or make the figure panels smaller. It is currently hard as it spreads to three pages.
As suggested by the reviewer, Fig 6 has been rearranged in only one page of the revised manuscript.
Fig 6A – The labeling for the domain needs to be clear. For example, ΔNTD+CTD seems like the construct has ΔNTD with CTD. This can be rectified by saying ΔNTD+ΔCTD or Δ(NTD+CTD).
As suggested by the reviewer, we used ΔNTD+ΔRBD, ΔNTD+ΔCTD, ΔRBD+ΔCTD to replace ΔNTD+RBD ΔNTD+CTD, ΔRBD+ CTD, respectively, in Fig 6A of the revised manuscript. Furthermore, the relevant modifications have been made on Page 2 lines 90-91 and Page 12 lines 287 and 289 of the revised manuscript.
